# Loss of Protein Function Causing Severe Phenotypes of Female-Restricted Wieacker Wolff Syndrome due to a Novel Nonsense Mutation in the *ZC4H2* Gene

**DOI:** 10.3390/genes13091558

**Published:** 2022-08-29

**Authors:** Jing-Jing Sun, Qin Cai, Miao Xu, Yan-Na Liu, Wan-Rui Li, Juan Li, Li Ma, Cheng Cai, Xiao-Hui Gong, Yi-Tao Zeng, Zhao-Rui Ren, Fanyi Zeng

**Affiliations:** 1Shanghai Institute of Medical Genetics, Shanghai Children’s Hospital, Shanghai Jiao Tong University School of Medicine, Shanghai 200040, China; 2Department of Neonatology, Shanghai Children’s Hospital, Shanghai Jiao Tong University School of Medicine, Shanghai 200062, China; 3Department of Histo-Embryology, Genetics and Developmental Biology, Shanghai Jiao Tong University School of Medicine, Shanghai 200025, China; 4NHC Key Laboratory of Medical Embryogenesis and Developmental Molecular Biology & Shanghai Key Laboratory of Embryo and Reproduction Engineering, Shanghai 200040, China

**Keywords:** WRWF, WRWFFR, ZC4H2, nonsense mutation

## Abstract

Pathogenic variants of zinc finger C4H2-type containing (*ZC4H2*) on the X chromosome cause a group of genetic diseases termed ZC4H2-associated rare disorders (ZARD), including Wieacker-Wolff Syndrome (WRWF) and Female-restricted Wieacker-Wolff Syndrome (WRWFFR). In the current study, a de novo c.352C>T (p.Gln118*) mutation in *ZC4H2* (NM_018684.4) was identified in a female neonate born with severe arthrogryposis multiplex congenita (AMC) and Pierre-Robin sequence (cleft palate and micrognathia). Plasmids containing the wild-type (WT), mutant-type (MT) *ZC4H2,* or *GFP* report gene (N) were transfected in 293T cell lines, respectively. RT-qPCR and western blot analysis showed that ZC4H2 protein could not be detected in the 293T cells transfected with MT *ZC4H2*. The RNA seq results revealed that the expression profile of the MT group was similar to that of the N group but differed significantly from the WT group, indicating that the c.352C>T mutation resulted in the loss of function of ZC4H2. Differentially expressed genes (DEGs) enrichment analysis showed that c.352C>T mutation inhibited the expression levels of a series of genes involved in the oxidative phosphorylation pathway. Subsequently, expression levels of *ZC4H2* were knocked down in neural stem cells (NSCs) derived from induced pluripotent stem cells (iPSCs) by lentiviral-expressed small hairpin RNAs (shRNAs) against *ZC4H2*. The results also demonstrated that decreasing the expression of *ZC4H2* significantly reduced the growth of NSCs by affecting the expression of genes related to the oxidative phosphorylation signaling pathway. Taken together, our results strongly suggest that *ZC4H2* c.352C>T (p.Gln118*) mutation resulted in the loss of protein function and caused WRWFFR.

## 1. Introduction

Wieacker-Wolff syndrome (WRWF; OMIM: 314580) was first reported as a severe X-linked neurodevelopmental disorder [1]. The male patients reported were all from big families with missense mutations passed on by female carriers. Male patients had early lethal phenotypes or were otherwise severely affected and had no offspring, whereas the female carriers were often asymptomatic [1]. The symptoms are primarily characterized by neurological abnormalities, including motor retardation, intellectual disability, brain atrophy, congenital general hypotonia, poor speech, dysphagia, and arthrogryposis multiplex congenita (AMC) associated with multiple dysmorphic features [2,3,4]. Short stature, muscle weakness, feeding difficulties, laryngomalacia, hypoglycemia, and apnea, among other symptoms, were also often reported [4].

The prevalence of WRWF is estimated to be <1 per 1,000,000 population [5]. The zinc finger C4H2-type containing (*ZC4H2*) gene located on chromosome Xq11.2 was confirmed to be the pathogenic gene underlying WRWF [2]. However, there has been an increase in the number of case reports showing a rise in the number of female patients with severe phenotypes [5,6,7,8,9,10,11], and non-skewed X chromosome inactivation (XCI) was reported in certain female patients [5,7,9]. In 2019, a total of 23 families and simplex cases were reported with *ZC4H2* defects, including 19 affected females in 18 families and 14 affected males in 9 families [4]. It was found that females with deleterious de novo *ZC4H2* variants (splicing, frameshift, nonsense, and partial deletions) presented with a mild to severe phenotype. Therefore, female-restricted Wieacker-Wolff Syndrome (WRWFFR; OMIM: 301041) is now considered an X-linked dominant syndrome. WRWF and WRWFFR are termed ZC4H2-associated rare disorders (ZARD) [4].

*ZC4H2* encoded protein contains a C-terminal zinc-finger domain (ZNF domain), a CC domain, and a putative nuclear localization signal (NLS) [9,12]. The zinc-finger protein is characterized by the presence of four cysteine residues and two histidine residues in the ZNF domain [13]. It was found that in mouse primary hippocampal neurons, transient produced ZC4H2 altered protein influenced dendritic spine density [2]. Analysis of cell-type-specific markers showed a specific loss of V2 interneurons in the brain and spinal cord in the *zc4h2* knockout zebrafish model [3]. It was recently identified that Zc4h2 was required for the development of central noradrenergic (NA) neurons in the mouse brain, which indicated that the disease affected the development of the nervous system [14].

The clinical phenotype varies widely among different ZARD patients. For example, microcephaly was observed in several patients. MRI analysis showed encephalatrophy, ventriculomegaly, and abnormal white matter myelination in certain patients. However, previous studies may not entirely explain the mechanism by which comprehensive developmental and intellectual disability occurs in ZARD patients.

In the present study, a novel potentially pathogenic nonsense mutation (c.352C>T) of *ZC4H2* (NM_018684.4) was identified in a female neonate who was suffering from AMC and Pierre-Robin sequence (cleft palate and micrognathia). The results of in vitro overexpression and knockdown experiments indicated that the resultant loss of protein function caused by the nonsense mutation may influence the oxidative phosphorylation pathway and may be involved in the pathogenicity of the female-restricted Wieacker Wolff Syndrome.

## 2. Materials and Methods

### 2.1. Ethical Compliance

The patient in this study was hospitalized in the Neonatology Department of Shanghai Children’s Hospital. The clinical evaluation was conducted in accordance with the principles of the Declaration of Helsinki [15]. The study was approved by the Ethics Committee of Shanghai Children’s Hospital, the Affiliated Children’s Hospital of Shanghai Jiao Tong University School of Medicine in China. Written informed consent was obtained from the parents of the patient.

### 2.2. Next-Generation Sequencing and Sanger Sequencing

Genomic DNA was extracted from peripheral blood lymphocytes from the patient and her parents. Trio clinical exome sequencing, which encompassed 4796 genes, was performed to identify potential pathogenic mutations. Coding exons of the gene panel were captured using a TruSight^®^ Rapid Capture kit (Illumina, Inc., San Diego, CA, USA) according to the manufacturer’s instructions and sequenced on a HiSeq 4500 platform (Illumina, Inc.) using pair-end reads.

Data processing, sequence alignment (to GRCh37), variant filtering, and ranking by allele frequency were subsequently performed. A proband-parents trio-based strategy was used for the analysis [16]. The variants were interpreted and categorized according to the five-tier classification system recommended by the American College of Medical Genetics and Genomics in 2015 [17]. The population frequency for the variant in this report was reviewed in the Genome Aggregation Database (gnomAD). Sanger sequencing was subsequently performed to confirm the de novo mutation in *ZC4H2*, as identified by next-generation sequencing. Primers were designed to cover the mutation sequence. Forward primer: 5′-TGCCTATCCCACTCTATGTTCC-3′ and reverse primer: 5′-TATACCTGCCCGTGTGTGTG-3′.

### 2.3. X-Chromosome Inactivation (XCI) Assay

DNA extracted from the proband and her parents were subsequently used for an XCI assay. One sample was subjected to methylation-sensitive endonuclease (*Hha*I) for digestion, whereas a second without endonuclease digestion was used as a control. The parents’ DNA was used as a control without endonuclease. The androgen receptor gene was subsequently amplified using fluorescently labeled primers and capillary electrophoresis, and the ratio of XCI in the proband was calculated according to the areas of two peaks inherited from his parents as described previously [18].

### 2.4. Domain and Secondary Structure Analysis of De Novo Nonsense Mutation

Domain analysis was performed using the InterPro [19] (ebi.ac.uk/interpro/) tool, which can recognize motifs and domains of a protein, making it possible to functionally characterize proteins using this database; the database consists of protein families, domains and functional sites.

Secondary structure, binding-analysis, and functional analysis were subsequently performed using the PredictProtein server [20], the first internet server developed for protein predictions that pioneered combining evolutionary information and machine learning. By providing a protein sequence as the input, the server outputs predictions such as secondary structure, solvent accessibility, and protein function.

### 2.5. Construction of the ZC4H2 Expression Vectors

The *ZC4H2* (NM_018684) Human Tagged ORF Clone with C-terminal Myc-DDK tag plasmid (Cat. no. RC202589) was purchased from OriGene Technologies, Inc. (Rockville, MD, USA) The *ZC4H2* c.352C>T mutation was generated using a ClonExpress^®^ Ultra One Step Cloning Kit (Cat. no. #C115; Vazyme Biotech Co., Ltd. (Nanjing, China)) with the following primers: forward, 5′-ATGACTCTGGGCCTGTAGAGGCTCCCTGACTTGTGTGA-3′ and reverse, 5′-TACAGGCCC AGAGTCATGCGCAGGGCATCCAC-3′. The same kit was used to construct the N-terminal Flag-tagged expression vector with wild-type *ZC4H2* (i.e., pCMV6-Flag-ZC4H2-wt) and mutant-type *ZC4H2* (i.e., pCMV6-Flag-ZC4H2-mt), respectively. The N-terminal Flag-tagged sequence before the open reading frame was ATGGATTACAAGGATGACGACGATAAG, and the N-terminal Flag-tagged expression vector was generated using the following primers: Forward, ttacaaggatgacgacgataagATGGCAGATGAGCAAGAAATCAT; and reverse, cgtcgtcatccttgtaatccat GGCGATCGCGGCGGCAGA; the upper and the lower case letters refer to the homologous sequences of the vector and Flag tag sequence to be imported, respectively. Four vectors were separated by agarose gel electrophoresis, and the sequences were confirmed by Sanger sequencing.

### 2.6. Cell Transfection and Immunoblot Analysis

293T cells were cultured in Gibco^®^ Dulbecco’s Modified Eagle medium (Thermo Fisher Scientific, Inc. (Waltham, MA, USA)) supplemented with 10% Gibco^®^ FBS (Thermo Fisher Scientific, Inc.) and incubated at 37 °C in a humidified incubator supplied with 5% CO_2_. Penicillin (100 units/mL) and streptomycin (100 μg/mL) were added to the culture media. 293T cells were transfected with pCMV6-Flag-ZC4H2-wt, pCMV6-Flag-ZC4H2-mt, and pLVHM-GFP (control vector), and these treatments comprised the experimental groups termed as the WT, MT, and N groups, respectively. The transfection experiments were performed using Invitrogen™ Lipofectamine^®^ 3000 (Thermo Fisher Scientific, Inc.) according to the manufacturer’s instructions. At 48 h after transfection, cells were lysed in RIPA lysis buffer (Strong; Beyotime Institute of Biotechnology (Suzhou, China)) and PMSF (final concentration: 1 mM, Beyotime Institute of Biotechnology), and centrifuged for 5 min at 18,000× *g* at 4 °C. RT-quantitative PCR (RT-qPCR) was used to measure the expression of WT type *ZC4H2* and MT *ZC4H2*. Primers were designed before the mutation to explore whether gene expression was altered. Forward primer, 5′-CTGAGTTTGAGGCACTTGAG-3′, and reverse primer, 5′-GATTGTTTGATAGTGTTTTCCATC-3′.

Immunoblots were incubated with the following primary and secondary antibodies: ZC4H2 polyclonal antibody (1:1000; Cat. no. PA5-48116, Invitrogen; Thermo Fisher Scientific, Inc.), FLAG tag rabbit polyclonal antibody (1:1000 Cat. no. AF0036), β-actin rabbit monoclonal antibody (1:1000; Cat. no. AF5003, both from Beyotime Institute of Biotechnology), and anti-rabbit IgG, HRP-linked antibody (1:10,000; Cat. no. #7074, Cell Signaling Technology, Inc. (Danvers, MA, USA)), respectively.

### 2.7. Lentivirus Vectors for ZC4H2 Small Hairpin RNA (shRNA)

Lentivirus shRNA transfer vectors with green fluorescent protein (GFP) sequence were constructed by Shanghai GeneChem Co., Ltd. (Shanghai, China). The sequences of the three pairs of shRNA used to target the human *ZC4H2* gene are listed in Appendix A, and successful cloning into plasmids was confirmed by sequencing. The recombinant lentivirus vector containing the shRNA targeting *ZC4H2* and the control lentivirus were prepared and titered to 1 × 10^8^ TU/mL (transfection unit).

### 2.8. Induced Differentiation of Neural Stem Cells (NSCs) and Lentivirus shRNA Gene Transfection

Normal induced pluripotent stem cells (iPSCs) were purchased from ATCC (ACS-1011, newborn, male). The iPSCs were cultured in Gibco™ StemFlex™ medium (Thermo Fisher Scientific, Cat. no. A3349401), and induction of differentiation of NSCs was performed according to the standard protocols of PSC Neural Induction Medium (Thermo Fisher Scientific, Cat. no. A1647801). After 21 days of induction, cells on coverslips were fixed in 4% formaldehyde for 30 min and permeabilized using 1× PBS containing 0.2% TritonX-100 on ice for 15 min. Then cells were incubated with blocking solution at room temperature for 60 min. Cells were incubated with the primary antibodies against SOX1 (R&D systems (Minneapolis, MN, USA), Cat. no. AF3369-SP), NESTIN (R&D systems, Cat. no. MAB1259-sp) and PAX6 (Abcam, Cat. no. ab195045) diluted in blocking buffer overnight at 4 °C. Subsequently cells were incubated with Alexa Fluor 488-labeled Goat Anti-Rabbit IgG (H+L) (Beyotime Institute of Biotechnology, Cat. no. A0423), Cy3-labeled Goat Anti-Mouse IgG (H+L) (Beyotime Institute of Biotechnology, Cat. no. A0521) and Alexa Fluor 555-labeled Donkey Anti-Rabbit IgG (H+L) (Beyotime Institute of Biotechnology, Cat. no. A0453) at 4 °C in the dark, which was followed by DAPI staining at room temperature for 8 min. Cells were imaged using a fluorescence microscope (Nikon Corporation (Tokyo, Japan), TE2000).

Induced NSCs were seeded in 6-well plates at a density of 0.5–1 × 10^5^ cells per cm^2^ and cultured in Neural Expansion Medium with Advanced™ DMEM⁄F-12 (Thermo Fisher Scientific, Cat. no. 12634028) at 37 °C in a humidified incubator supplied with 5% CO_2_. The following day, after removing the culture medium, lentiviruses were inoculated into cells at a multiplicity of infection (MOI) of 12. The efficiency of infection (72 h after infection) was monitored under a fluorescent microscope to evaluate GFP expression. The expression levels of *ZC4H2* were measured using RT-qPCR in the NSCs transfected with a lentivirus containing one of the three different shRNAs.

After removing the lentiviruses, NSCs were incubated with 2 μg/mL puromycin (Sigma-Aldrich (St. Louis, MO, USA); Merck KGaA) in Neural Expansion Medium with Advanced™ DMEM⁄F-12 for >10 days. The status of NSCs was observed by fluorescence microscope.

### 2.9. RNA Sequencing (RNA-Seq) and Processing RNA Seq Data

Total RNA was extracted from 293T cells or NSCs using TRIzol™ (Tiangen Biotech Co., Ltd. (Beijing, China)) reagent, and RNA purity was assessed using an Implen NanoPhotometer^®^ spectrophotometer (Implen USA, Inc. (Village, CA, USA)). RNA integrity was assessed using the RNA Nano6000 Assay Kit of the Bioanalyzer 2100 system (Agilent Technologies, Inc. (Santa Clara, CA, USA)).

A total of 3 µg RNA per sample was used as input material for the RNA sample preparation. Sequencing libraries were generated using NEBNext^®^ Ultra™ RNA Library Prep kit for Illumina^®^ (New England BioLabs, Inc. (Ipswich, MA, USA)) following the manufacturer’s recommendations, and index codes were added to attribute sequences to each sample. The library preparations were sequenced on a NovaSeq 6000 platform (Illumina, Inc.), and 125 bp/150 bp paired-end reads were generated.

The raw reads were first processed to remove any reads that contained adapter sequences and those containing poly-N and low-quality reads. After filtering the reads, the index of the reference genome was built using HISAT2 (version 2.0.5, The Center for Computational Biology, Johns Hopkins University, MD, USA), and paired-end clean reads were aligned to the reference genome. 

FeatureCounts (version v1.5.0 p3, The Walter and Eliza Hall Institute of Medical Research, Parkville, Australia) was used to count the read numbers mapped to each gene. Subsequently, the fragments per kilobase of transcript per million mapped reads of each gene were calculated based on the length of the gene in question and the reads count mapped to this gene.

Differential expression analysis of each group was performed using the DESeq2 (version 1.16.1, European Molecular Biology Laboratory, Heidelberg, Germany). The resulting *p*-values were adjusted using the Benjamini and Hochberg’s approach for controlling the false discovery rate. Genes with an adjusted *p*-value < 0.05 identified by DESeq2 were assigned as differentially expressed genes (DEGs).

Gene Ontology (GO) and Kyoto Encyclopedia of Genes and Genomes (KEGG) pathway enrichment analysis of DEGs were performed using the R package clusterProfiler [20]. *p* < 0.05 was considered to indicate a significant difference for a DEG.

### 2.10. PCR Array Analysis

The genes involved in mitochondrial molecular transport were evaluated using the RT^2^ Profiler PCR Array analysis (human mitochondria; PAHS 087ZA, Qiagen GmbH (Hilden, Germany)); genes required for maintaining the mitochondrial membrane polarization were potentially crucial for ATP synthesis were assessed. Each cDNA sample was diluted with nuclease-free water and mixed with RT2 SYBR™ green MasterMix (Qiagen GmbH). RT-qPCR was performed using the following thermocycling conditions: 95 °C for 10 min; followed by 40 cycles of 95 °C for 15 s and 60 °C for 30 s. Data were analyzed using the Qiagen RT2 Profiler PCR Array Data Analysis Web Portal. The fold difference between groups was calculated using the 2^−ΔΔCt^ methods [21].

### 2.11. RT-qPCR

After total RNA was extracted from 293T cells or NSCs, cDNA was synthesized using a HiScript II Q RT SuperMix for qPCR (+gDNA wiper) kit (Vazyme Biotech Co., Ltd.) according to the manufacturer’s instructions. RT-qPCR was performed using specific TaqMan^®^ probes (Appendix A) and the PCR MasterMix (Thermo Fisher Scientific, Cat. no. 10572014) or specific primers (Appendix A) and FastStart PCR Master (Roche Diagnostics, Inc., Cat. no. 4710436001) on an Applied Biosystems 7500 Fast Real-Time PCR System (ABI 7500; Thermo Fisher Scientific, Inc.), respectively. The thermocycling conditions used were: 95 °C for 2 min; followed by 40 cycles of 95 °C for 30 s and 60 °C for 30 s. The relative expression levels were calculated using the 2^−ΔΔCt^ method and normalized against the levels of GAPDH.

## 3. Results

### 3.1. Presentation of Clinical Case

The patient was the first child of healthy nonconsanguineous parents of Chinese origin, and there was no prior family history of known inherited disorders. Cesarean section delivery was performed due to fetal distress at 36^+5^ weeks of gestation. The total Apgar score was 10 at 1 and 5 min after birth, which was normal. The female newborn was found to have multiple abnormalities and admitted to our neonatal ward 1 h after birth. After admission, the baby’s body weight and frontal occipital circumference were measured at 2655 g and 32.5 cm, respectively; both were slightly lower than the 50th percentile. The exact body length could not be adequately determined because of the profound abnormalities. 

Patients with severe WRWFFR can manifest multiple neurological abnormalities with various dysmorphic features. In this study, the neonate was found to display typical clinical characteristics of severe WRWFFR, such as AMC and congenital developmental malformations (Figure 1A–F), as well as feeding difficulties. Radiological findings confirmed these clinical characteristics (Figure 1G–L). The cranial ultrasound revealed a left subependymal iso echoic structure of 9.3 × 3.5 mm, and on the right 8.1 × 4.1 mm, which suggested bilateral subependymal hemorrhage. An enlarged posterior horn of the left lateral ventricle was also identified (anterior horn, 2.6 mm; the body, 4.6 mm; and the posterior horn, 12.2 mm). Neonatal screening with blood tandem mass spectrometry using bloodspot did not display additional abnormalities. Unfortunately, the EMG/ENG, echocardiography, electrocardiogram, and electroencephalogram were not able to be obtained in time before the proband died two weeks after birth. Next-generation sequencing was subsequently performed to identify the genetic abnormality and to explore the possible mechanism underlying this disease.

### 3.2. Identification of the De Novo Nonsense Mutation (c.352C>T) in the ZC4H2 Gene

A de novo *ZC4H2* (NM_018684.3, exon 3 c.352C>T) mutation was identified in the proband by next-generation sequencing and validated by Sanger sequencing (Figure 2A). The variant was classified as pathogenic according to the American College of Medical Genetics and Genomics (ACMG) guidelines [17], and it had not been previously described in the gnomAD database [22]. The results of the XCI assay showed that the ratio of the expressing paternal X chromosome to the maternal chromosome was 59:41 in the proband, which was consistent with the random pattern (Figure 2B). Secondary structure, solvent accessibility, and domain analysis, as well as binding residues predictions, were performed using PredictProtein, InterPro, and ProNA2020, respectively [23,24]. The results revealed that the ZC4H2 protein had a 110-amino-acid-long helix, and most residues were exposed to solvents (Appendix A). The nonsense mutation caused the protein to lose its C-terminal zinc-finger domain, together with the DNA-binding residues. However, the 110-amino-acid-long helix and specific protein binding residues were retained in the truncated protein.

### 3.3. In Vitro Analysis Showing the Uncoupling of Transcription and Translation of the Mutant Gene in 293T Cells

Next, 293T cells were transfected with three different expression vectors, including pCMV6-Flag-ZC4H2-wt, pCMV6-Flag-ZC4H2-mt, and pLVHM-GFP, respectively (Figure 2C). The RT-qPCR analysis results revealed no significant differences in the mRNA expression of *ZC4H2* between the MT and WT groups (*p* = 0.72; Figure 2D). It was interesting to note that the ZC4H2 protein could be detected in the WT group. However, it could not be detected in the MT group, either with the ZC4H2 polyclonal antibody or Flag antibody (Figure 2E). Therefore, it is speculated that the truncated protein may be unstable in transfected 293T cells.

### 3.4. Global Transcription Profiling Using RNA-seq and Prediction of Aberrantly Regulated Pathways in the Transfected 293T Cells

RNA-seq generated 401 million raw reads, in which 390 million clean reads with an average of 43 million clean reads per sample were obtained in 293T cells. The RNA-seq results showed that the expression pattern in the MT group was similar to that in the N group but significantly different from the WT group (Figure 3A). A total of 1618 DEGs (Benjamini & Hochberg adjusted *p* < 0.05; |log2FoldChange| > 0.0) were detected between the WT and MT groups, of which 847 genes were up-regulated and 771 genes were down-regulated in the MT group. In addition, 723 DEGs (including 351 up-regulated genes and 372 down-regulated genes) were identified between the WT and N groups, and 234 DEGs were identified between the MT and N groups (including 110 up-regulated genes and 124 down-regulated genes). The results showed that a series of genes involved in oxidative phosphorylation, Parkinson’s disease, and the Thermogenesis pathway were overexpressed in the WT group compared with the MT or the N groups (Figure 3B). After further analysis of the PCR array and RT-qPCR, it was found that the DEGs, which were related to mitochondrial function, were overexpressed in the WT group (Figure 3C–G). These results confirmed that the c.352C>T mutation of the *ZC4H2* gene could affect the expression of the oxidative phosphorylation complex, thus affecting mitochondrial function in 293T cells.

### 3.5. Inhibition of ZC4H2 Expression Severely Affects the Growth of NSCs

After induction for 10 days, iPSCs were differentiated to be NSCs. Some critical markers of neural differentiation including NESTIN, PAX6, and SOX1 were highly expressed in NSCs (Appendix A). The interference efficiency of shRNA-mediated knockdown was identified. The results of the RT-qPCR analysis showed that *ZC4H2* expression levels in each RNAi group of cells were significantly lower compared with the negative control group cells (*p* < 0.01, Appendix A). As shown in Appendix A, ZC4H2-shRNA-1 (S1) was found to be the most effective inhibitor, with an inhibition rate of 75%. Therefore, the NSCs infected with lentivirus S1 were chosen for the subsequent RNA-seq experiments.

The lentivirus infection efficiency was monitored based on GFP expression. Over 90% of the cells were successfully infected after three days of transfection. Subsequently, Puromycin was added. On day 1 and day 10, significant cell proliferation and normal cell morphology were observed. In contrast, the growth of NSCs was severely inhibited after they were infected with the *ZC4H2* shRNA lentivirus, and the inhibition was negatively correlated with the expression levels of *ZC4H2* in the NSCs. On day 10, after infection with ZC4H2-shRNA-1 or ZC4H2-shRNA-3, almost all of the NSCs had died (Figure 4A); only one cluster of NSCs could be observed after infection with ZC4H2-shRNA-2, in which the expression levels of *ZC4H2* were greater than half of that of the normal controls (Appendix A). These results suggest that *ZC4H2* inhibition prevented the survival and growth of NSCs.

### 3.6. Differential Expression of Genes and Pathways in the NSCs following ZC4H2 Knockdown

RNA-seq generated 262.5 million raw reads, in which 240.8 million clean reads with an average of 40.1 million clean reads per sample were obtained in the NSCs. A total of 4091 DEGs (Benjamini & Hochberg adjusted *p* < 0.05; |log2FoldChange| > 0.0) were detected between the ZC4H2-shRNA group (S group) and normal controls (N group), of which 1608 genes were up-regulated and 2483 genes were down-regulated in the S group (Figure 4B,C). The KEGG pathway enrichment assay further determined the signaling pathways involved in DEGs. The bubble map of the differential gene provides a graphical representation of the top 15 most enriched pathways in DEGs (Figure 4D). Among these enriched signaling pathways, oxidative phosphorylation, Parkinson’s disease, Alzheimer’s disease, and Huntington’s disease were the most significantly downregulated in the S group. These enriched KEGG pathways from the differentially expressed genes are listed in Appendix A. Then, venny diagram analysis was performed for the DEGs involved in the above pathways. The results indicated that a total of 44 genes were involved in these pathways, which play important roles in the oxidative phosphorylation pathways. As shown in Figure 4E,F, these significantly down-regulated genes are primarily located on the oxidative phosphorylation complex I (NDUFA1, NDUFA3, NDUFB3, etc.) and complex IV (COX6A1, COX6B1, COX8A, etc.).

## 4. Discussion

ZARD displays a broad phenotypic spectrum, caused by mutations in the *ZC4H2* gene [3,25]. Severe symptoms and even lethal phenotypes can manifest in both males and females, while mild or no symptoms are typically only observed in female patients. The various symptoms appear to correlate well with the different mutations in the *ZC4H2* gene that have been reported [3,4,5,7,8,9,10,26,27,28].

The human *ZC4H2* gene is located on the long arm of the X chromosome (Xq11.2). Previous study has confirmed that *ZC4H2* is subject to X inactivation in females and the XCI pattern affects *ZC4H2* expression in females [4]. Thus, both the XCI patterns and the nature of the mutations are thought to affect the different disease states [29]. Hirata et al. observed skewed XCI and preferential inactivation of the mutated X chromosome in the females tested, with the exception of one affected girl and her unaffected mother [2]. Now it has become clear that WRWFFR is an X-linked dominant disorder, and de novo *ZC4H2* deletions and splicing, frameshift, and nonsense mutations could result in severe phenotypes in female heterozygotes without skewed XCI [4,5,7,9]. To date, six female patients of WRWFFR with four de novo nonsense mutations have been identified. A novel de novo nonsense mutation (c.352C>T) was confirmed in the patient of the present study. A severe phenotype, with AMC, digital deviations, and dysmorphic features including short neck and Pierre–Robin sequence (cleft palate and micrognathia), resembling that of the phenotype from other female patients with nonsense mutations [4,6], was also displayed, while other clinical symptoms, such as subependimal hemorrhage, were likely due to complication of preterm birth. In this patient, the XCI assay showed that the ratio of the expressing paternal X chromosome to the maternal chromosome was 59:41, which was consistent with the random pattern. Accordingly, to the best of our knowledge, the novel de novo nonsense mutation (c.352C>T) in *ZC4H2* identified correlated with the severe clinical phenotypes presented.

Previous reports showed that the C-terminal domains were important for the function of ZC4H2, through binding to other proteins and factors. The majority of the mutations identified to date are located in the C-terminal region of the ZC4H2 protein, including both the ZNF domains and the NLS [9,12]. In this study, a c.352C>T mutation was predicted to produce a truncated protein containing only 117 amino acids in length, resulting in the loss of the ZNF domains and the NLS. And thus, most likely, crucial functions, such as nervous system development and the regulation of neuron differentiation, would be lost when the truncated protein is expressed.

To further investigate the pathological origin at the protein level, in vitro cell transfection experiments using wild-type *ZC4H2* and mutant-type *ZC4H2* expression vectors were performed with 293T cells. Unexpectedly, the predicted truncated protein could not be identified in the 293T cells transfected with the mutant vector. However, no significant differences in the mRNA expression of *ZC4H2* were observed in the 293T cells transfected with wild-type vector or the mutant vectors, suggesting that the truncated protein was unstable or rapidly degraded in the cells. This in turn could explain the severity of the disease symptoms in the patient.

The RNA-seq results showed that the loss of ZC4H2 protein function caused by c.352C>T mutation had a negative effect on the oxidative phosphorylation pathway in 293T cells. Therefore, we further investigated the effect of *ZC4H2* expression on the proliferation of NSCs. Interestingly, inhibition of *ZC4H2* gene expression was closely related to the survival and growth of NSCs, and excessive inhibition of the *ZC4H2* gene expression resulted in the death of NSCs.

Mitochondria are signaling hubs responsible for the generation of energy through oxidative phosphorylation. Over the last decade, an increasing body of data has highlighted the fact that oxidative phosphorylation is essential for cells to meet the high-energy demands of differentiation. The inhibition of mitochondrial respiration due to oxidative phosphorylation pathway dysfunction has been shown to lead to the impairment of neuronal differentiation [30]. Emerging evidence indicates that mitochondria are central regulators of NSCs fate decisions and are crucial for neurodevelopment [31]. Our results also confirmed that the expression of the oxidative phosphorylation complex, especially complex I and complex IV, was significantly down-regulated in the NSCs following lentiviral vector-mediated *ZC4H2* knockdown.

Cytochrome c oxidase (COX) is complex IV of the respiratory chain and it catalyzes the transfer of electrons from cytochrome c to oxygen coupled to proton pumping from the mitochondrial matrix to the intermembrane space. Previous reports indicate that loss of function of the oxidative phosphorylation complex I and complex IV play critical roles in nervous system diseases [32,33,34,35,36].

It is an interesting question how oxidative phosphorylation disruption contributes to the clinical phenotype of WRWFFR. Many birth defects, such as Miller syndrome, are associated with craniofacial malformations [37]. Miller syndrome is a type of acrofacial dysostosis, caused by the mutation of the DHODH gene located on 16q22. It was reported that DHODH depletion partially inhibits respiratory chain complex III activity and increases the generation of ROS [37], and the development of neural crest stem cells is involved in craniofacial malformations [38]. The results suggested that failure of neural crest cell development caused by oxidative phosphorylation disruption may be a contributing factor to clinical symptoms such as cleft palate and micrognathia, similar to WRWFFR.

In conclusion, we can speculate that loss of protein function due to a novel nonsense (c.352C>T) mutation in the *ZC4H2* gene may lead to severe phenotypes of WRWFFR by down-regulating the expression level of the oxidative phosphorylation complex. Further exploration of mitochondrial function as well as the role played in the patients with the down-regulation of oxidative phosphorylation pathway is warranted and may provide novel avenues in the approaches for treating WRWFFR.

## 5. Conclusions

A novel nonsense mutation of *ZC4H2* c.352C>T (p.Gln118*) was identified in a female patient suffering from WRWFFR, and this mutation was shown to lead to the loss of protein function of the *ZC4H2* gene. Cellular functional analysis showed the involvement of the aberrant oxidative phosphorylation pathway activity in the pathogenicity of this disease. The association between changes in pathway regulation and the clinical phenotypes requires further study.

## Figures and Tables

**Figure 1 genes-13-01558-f001:**
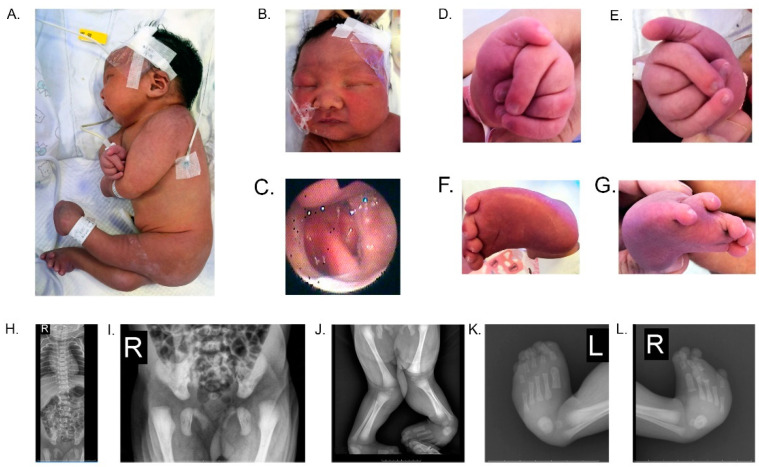
Clinical features of the female proband. Images of the patient 4 days after birth revealed: (**A**) a short neck, and AMC including shoulders in adduction, elbows in flexion, hip joints in flexion, knees in extension, radial and ulnar digital deviation, severe congenital club-feet, and limb spasticity, together with laying naturally on her side; (**B**) hypertelorism, micrognathia, and retrognathia; (**C**) cleft palate with a split of the uvula and soft palate, and a partial split of the hard palate, as observed by laryngoscopy; (**D**) radial and ulnar digital deviation of the left hand; (**E**) radial and ulnar digital deviation of the right hand; (**F**) severe left club foot; and (**G**) severe right club foot and ectopic fourth toe. At 3 days, the X-ray analysis showed: (**H**) no apparent bone abnormalities in the spine; (**I**) bilateral hip dysplasia; (**J**) X-shaped deformity of both lower extremities, end-swelling of both femurs, tibiae, and fibulae with increased bone density; (**K**) left varus foot and increased calcaneal bone density; and (**L**) right varus feet, increased calcaneal bone density and an ectopic fourth toe.

**Figure 2 genes-13-01558-f002:**
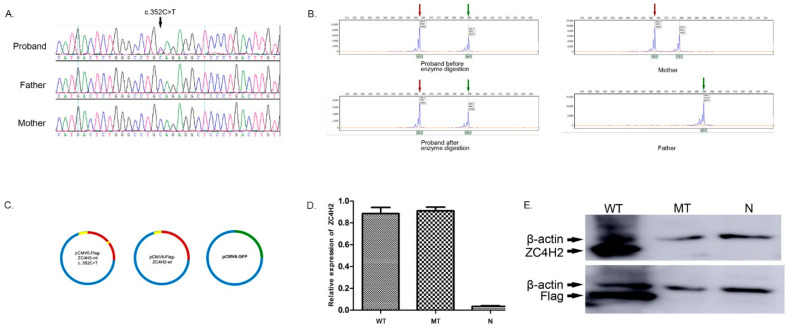
Characterization of the *ZC4H2* de novo mutation (c.352C>T). (**A**) Identification of the de novo mutation with Sanger sequencing. The c.352C>T mutation in exon 3 of *ZC4H2* was a de novo mutation identified only in the proband and not in her parents. (**B**) Analysis of X-chromosome inactivation. Two peaks of 262 and 294 bp were defined in the proband, indicating those inherited from her mother and father. The area under the peak inherited from the proband’s parents before enzyme digestion were 28,157 and 44,595, while those after enzyme digestion were 25,589 and 36,565, respectively. (**C**) Construction of the expression vectors. Wild-type and mutant proteins of ZC4H2 could be expressed in pCMV6-Flag-ZC4H2-wt and pCMV6-Flag-ZC4H2-mt c. 352C>T. The tagged protein (Flag) was inserted into the vectors, and the fusion protein was identified with antibodies targeting either Flag or ZC4H2. GFP was expressed by the normal control vector pLVHM-GFP, and therefore the transfection efficiency was determined by examining the fluorescence intensity. (**D**) Identification of relative expression levels of *ZC4H2* by RT-qPCR. The relative expression levels of *ZC4H2* were found to be 0.88 ± 0.058 and 0.91 ± 0.034, respectively. No significant differences were therefore identified between the two groups (*p* = 0.72). (**E**) Identification of the ZC4H2 protein in the 293T cells. Western blotting results showed that ZC4H2 and Flag could be detected in the cells transfected with pCMV6-Flag-ZC4H2-wt, whereas these were not seen in the cells transfected with pLVHM-GFP or pCMV6-Flag-ZC4H2-mt c. 352C>T. By contrast, β-actin was detected in all transfected cells.

**Figure 3 genes-13-01558-f003:**
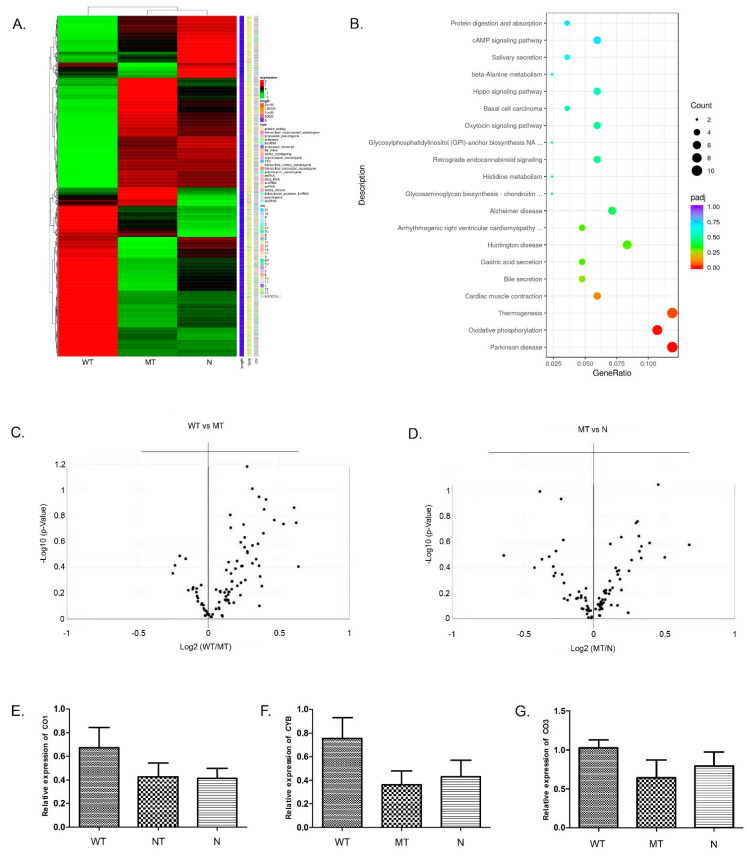
Gene expression profile of 293T cells with a de novo mutation (c.352C>T). (**A**) Heatmap of RNA-seq. A differential gene set of 2181 genes was obtained when the differential genes of all comparison groups were combined. The results of RNA-seq showed that the expression pattern in HEK293T cells transfected with the mutant was similar to that in the normal controls but significantly differed from that in the wild-type transfected cells. (**B**) KEGG pathway analysis of the up-regulated genes. The size and color of the circles correlate with the number of DEGs and the -log10 (*p*-value), annotated in the specific KEGG pathway, respectively. (**C**,**D**) PCR-array analysis for the genes related to the oxidative phosphorylation complex. The results indicated that most of the genes were up-regulated in the WT group. In contrast, gene expression patterns were very similar between the MT and N groups. The differentially expressed genes were normally distributed. (**E**–**G**) Verification of differentially expressed genes. The expression levels of some genes encoded by mitochondrial DNA, such as COX1, COX3, and CYB in the WT group, were higher than that in the MT and N groups, while the expression levels between the MT group and N group did not differ significantly.

**Figure 4 genes-13-01558-f004:**
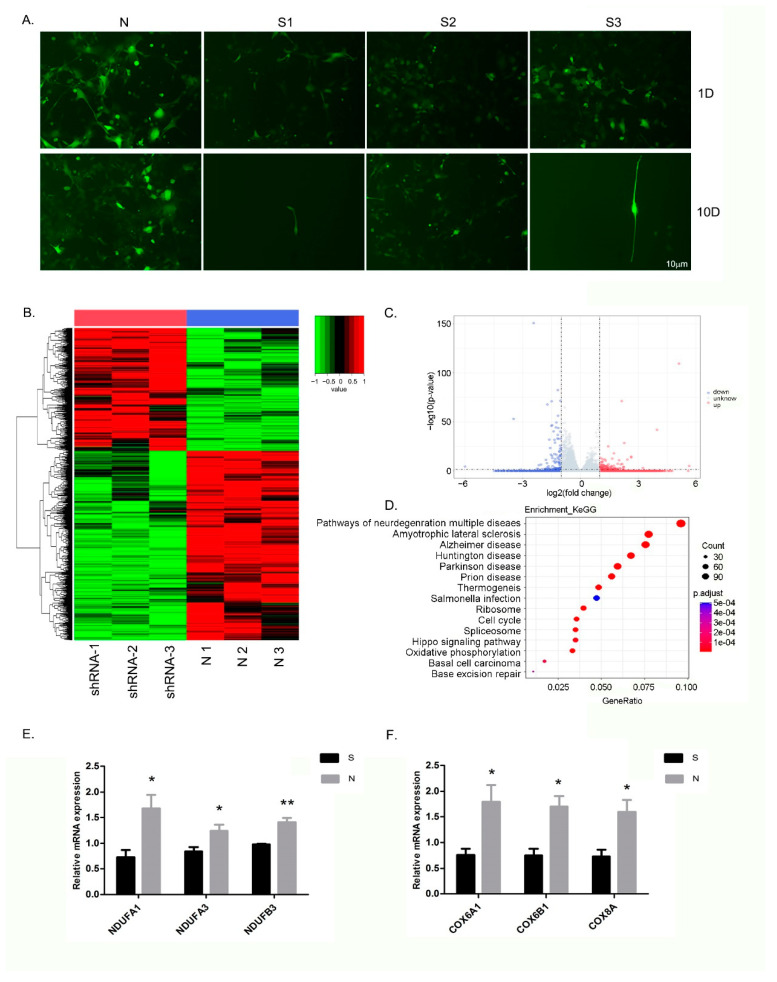
Effect of *ZC4H2* knockdown on the NSCs. (**A**) Growth of NSCs infected with lentiviral vectors at different points. N: the NSCs infected with the normal control lentiviruses (with a GFP reporter gene); S1–S3: the NSCs infected with three different shRNA-containing lentiviruses targeting the *ZC4H2* gene, respectively. The NSCs images were taken using a fluorescence microscope on day 1 and day 10 of selection using Puromycin. (**B**) Heatmap of RNA-seq. The NSCs infected with the shRNA-lentivirus targeting *ZC4H2* and normal controls, respectively. The experiments were repeated three times for each group. (**C**) Volcano plot of RNA-seq. A total of 4091 DEGs were obtained and 1608 genes were shown to be up-regulated and 2483 genes were down-regulated in the S group. (**D**) KEGG pathway analysis of the up-regulated genes. The top 15 significant KEGG pathways were shown. (**E**,**F**) RT-qPCR analysis for DEGs related to oxidative phosphorylation complex. * indicated significant differences compared with the control group (*p* < 0.05). ** indicates highly significant differences compared with the control group (*p* < 0.01).

## Data Availability

The data that support the findings of this study have been submitted to the Datesets are openly available at https://trace.ncbi.nlm.nih.gov/Traces/index.html?view=study&acc=SRP352771 (accessed on 28 December 2021) and https://www.ncbi.nlm.nih.gov/geo/query/acc.cgi?acc=GSE208171 (accessed on 17 July 2022). The reference number will be deposited when this paper is accepted.

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
