# Peer review of "Loss of Protein Function Causing Severe Phenotypes of Female-Restricted Wieacker Wolff Syndrome due to a Novel Nonsense Mutation in the ZC4H2 Gene"

_genes, 2022, doi:10.3390/genes13091558_

Round 1

Reviewer 1 Report

Overview:  The authors of this manuscript identified a new nonsense variant (p.Gln118*, c.352C>T) in ZC4H2 gene in a patient with Female-restricted Wieacker-Wolff Syndrome (WRWFFR). Overexpression of wildtype and mutant ZC4H2 in HEK293T cells indicated that the p.Gln118* variant caused degradation of the mutant ZC4H2 protein but had no effect on the ZC4H2 mRNA expression.  KEGG pathway analysis of RNA seq on ZC4H2 overexpressed HEK293T cells demonstrated that the oxidative phosphorylation signaling pathway was among the top enriched pathways. Specifically, expression of COX1, COX3 and CYB genes encoded by mitochondrial DNA were down regulated by the ZC4H2 p.Gln118* variant. Further,  functional analysis using iPSCs indicated that the knockdown of ZC4H2 inhibited neural stem cell growth. This study provided evidence of a new pathogenic ZC4H2 variant to the pool of WRWFFR causative variants and indicated that the oxidative phosphorylation signaling pathway is one of the major targeted pathways involved in the pathogenicity of WRWFFR.

Concerns:

1. Figure 2D. Plasmid overexpression in HEK293T cell line may not reflect the actual situation in the patient’s body. It is well known that nonsense variants in individuals usually result in nonsense mediated mRNA decay (NMD), which is a surveillance pathway that exists in all eukaryotes. It would be better to check the ZC4H2 mRNA expression in blood, tissues or cells that were derived from the patient if it is possible.

2. Figure 2D, it is difficult to read the height of the peak. The authors mentioned the area under the peak, but no data/number was provided. It would be help if the authors could describe the height number and area number in main text and perform a statistic analysis.

3. Figure 3A, the label of the color on the right side is difficult to read.

4. Page 12, line 401, “DESeq2 P<0.05; |log2FoldChange|>0.0”, did the authors performed multiple test correction for the RNAseq data? 

5. Figure 4A, a higher magnification (eg, 40x or 60x) image should be taken and presented as the current image was not of sufficient quality.

6. A discussion of how oxidative phosphorylation disruption contribute to arthrogryposis multiplex congenita (AMC) and Pierre-Robin sequence (cleft palate and micrognathia) could better connect the genotype and phenotype in this study.

Reviewer 2 Report

This article is well written and highly interesting, especially because it introduces links between mithocondrial dysfunction and ZC4H2 nonsense mutations (in addition to describing the effects of a novel one), paving the way for a better comprehension of molecular mechanisms underlying Wieacker Wolff Syndrome. Some major and minor revisions are required.

Major revisions:

  1. Introduction should be enriched by a deeper clinical description of the syndrome and by a summary of clinical and genetics data previously reported by other studies.

  2. Introduction, page 2, line 48-49, "female carriers are typically asymptomatic or mildly affected": this sentence sounds unclear, especially considering the following ones.

  3. Introduction, page 2, line 73: the term "mental retardation" must be substituted with "intellectual disability".

  4. Results, page 6, line 257-259: authors should avoid referring here to other patients and focus on theirs. Furthermore, if they refer to neurologic anomalies, authors should clearly describe them. In particular, authors should specify if signs of central and peripheral involvement are present, also in the light of the functional results discussed. In addition, authors should report on instrumental exams such as EMG/ENG, echocardiography, electrocardiogram, electroencephalogram, hormone dosages etc.

  5. Results, page 6, line 263-264, authors should specify what the "Genetic testing and neonatal screening with either blood tandem mass spectrometry or newborn bloodspot screening" searched for. The sentence is excessively vague in the current form.

  6. Was the child followed up after the neonatal period? Had neurologic signs been subsequently modifying?

  7. Discussion, page 12, line 423-424, the statement "ZC4H2 undergoes X inactivation in females" should be revised in the light of other findings by May et al., 2015; Zanzottera et al., 2017; Wang et al., 2020.

  8. Discussion, page 12, line 429-431: authors should rewrite these sentences in a more consistent way. In particular, they should discuss the statement "The XCI assay showed a random pattern" in comparison with other data reported in the literature regarding XCI in WRWFFR.

  9. Discussion, page 13, line 464, Cytochrome c oxidase is the last complex of the respiratory chain and it is the IV complex, not the IX.

  10. Discussion, page 13, line 466-469, "The phenotype of the patient in this study, most closely resembled the severe encephalopathy described for patients with mutations of the NDUFA4 and COX6B1 genes as well as a Leigh-like syndrome associated with leukodystrophy and severe epilepsy caused by a COX8A point-mutation": in the clinical description neither white matter anomalies resembling Leigh-like syndrome nor epilepsy have been reported. If the patient displayed these clinical symptoms, authors should describe them, otherwise this sentence should be erased.

Minor revisions:

  1. Abstract, page 1, line 23: please write "arthrogryposis multiplex congenita" in extenso before AMC.

  2. Abstract, page 1, line 29: please write "differentially expressed genes" in extenso before DEGs.

  3. Abstract, page 1, line 32: please write "small hairpin RNA" in extenso before shRNA.

  4. Keywords: Female-restricted Wieacker-Wolff Syndrome should be abbreviated as WRWFFR while Wieacker-Wolff Syndrome should be WRWF.

  5. Introduction, page 2, line 70: please substitute "microcephalus" with "microcephaly".

  6. Material and Method, page 4, line 171: please write "Neural Stem Cells" in extenso before NSCs.

  7. Figure 1, page 7: clubbed feet should be club-feet.

Round 2

Reviewer 2 Report

Major and minor revisions have been addressed and satisfied.